# Contrasting epigenetic control of transgenes and endogenous genes promotes post-transcriptional transgene silencing in *Arabidopsis*

Nicolas Butel [1,2,5], Agnès Yu[1,5], Ivan Le Masson[1], Filipe Borges [1], Taline Elmayan[1], Christelle Taochy[1,3], Nial R. Gursanscky [3], Jiangling Cao[3], Shengnan Bi [3], Anne Sawyer[3,4], Bernard J. Carroll [3,6✉] & Hervé Vaucheret[1,6✉]

Transgenes that are stably expressed in plant genomes over many generations could be assumed to behave epigenetically the same as endogenous genes. Here, we report that whereas the histone H3K9me2 demethylase IBM1, but not the histone H3K4me3 demethylase JMJ14, counteracts DNA methylation of *Arabidopsis* endogenous genes, JMJ14, but not IBM1, counteracts DNA methylation of expressed transgenes. Additionally, JMJ14-mediated specific attenuation of transgene DNA methylation enhances the production of aberrant RNAs that readily induce systemic post-transcriptional transgene silencing (PTGS). Thus, the JMJ14 chromatin modifying complex maintains expressed transgenes in a probationary state of susceptibility to PTGS, suggesting that the host plant genome does not immediately accept expressed transgenes as being epigenetically the same as endogenous genes.

[1] Institut Jean-Pierre Bourgin, UMR 1318, INRAE, AgroParisTech, Université Paris-Saclay, Versailles, France. [2] Université Paris-Sud, Université Paris-Saclay, Orsay, France. [3] School of Chemistry and Molecular Biosciences, The University of Queensland, Brisbane, QLD, Australia. [4] Queensland Alliance for Agriculture and Food Innovation, The University of Queensland, Brisbane, QLD, Australia. [5] These authors contributed equally: Nicolas Butel, Agnès Yu. [6] These authors jointly supervised this work: Bernard J. Carroll, Hervé Vaucheret. ✉email: b.carroll@uq.edu.au; herve.vaucheret@inrae.fr

RNA-mediated gene silencing is a highly conserved eukaryotic mechanism that regulates endogenous gene expression and acts as a defense mechanism against viruses and transposons[1–3]. It occurs when double-stranded RNA (dsRNA) is formed by transcription of an inverted repeat, the copying of single-stranded RNA (ssRNA) by viral and host RNA-dependent RNA polymerases (RDRs), or the simultaneous transcription of both the positive and negative strands of DNA. DICER-like (DCL) proteins convert the dsRNA into microRNAs (miRNAs) or small-interfering RNAs (siRNAs), which then guide ARGONAUTE-like (AGO) proteins to mediate sequence-specific gene silencing.

In plants, 24 nucleotide (nt) siRNAs guide transcriptional gene silencing (TGS) and RNA-directed DNA methylation (RdDM)[4–6]. The process of de novo RdDM also requires the cytosine methyltransferase DRM2[7]. Following the establishment of DNA methylation and through subsequent rounds of DNA replication, cytosine methylation at CG and CHG sites can be maintained in the absence of siRNA by the cytosine methyltransferases MET1, and CMT2 and CMT3, respectively, whereas maintenance of CHH methylation requires the continual presence of 24 nt siRNAs and DRM2[7,8].

Abundant siRNAs are also derived from viral RNA and RNA Polymerase II (Pol II) transcripts that have been converted into dsRNA by viral or host RDRs. In plants, RDR6 plays the major role in producing these dsRNAs, which are processed by DCL4 and DCL2 into 21 nt and 22 nt siRNAs, respectively. These siRNAs form a complex with AGO1 and guide silencing of complementary RNA viruses and post-transcriptional gene silencing (PTGS) of complementary mRNAs[9]. Once PTGS is triggered, the biogenesis of 21 and 22 nt siRNAs usually extends along the entire length of the mRNA in a process known as transitivity[10–12]. Furthermore, once induced, PTGS moves systemically throughout the plant[2,13,14].

It is well established that over-expression of aberrant mRNAs lacking a 5′ cap[15,16] or poly(A) tail[17] induces RDR6-dependent PTGS in plants. The transcription of aberrant RNAs is a by-product of RNA Pol II transcription in all eukaryotes, but normally, aberrant RNAs are intercepted and removed by the highly conserved RNA Quality Control (RQC) pathway. In addition to removing aberrant RNAs, the RQC pathway modulates gene expression via mRNA turnover. The mechanism of mRNA turnover has been studied in much more detail than the fate of aberrantly transcribed RNA, and the first step in mRNA decay is the removal of the polyA tail, followed by the removal of the 5′ cap by the decapping complex, which is composed of DCP1, DCP2 and VARICOSE (VCS) in plants[18]. Once the stabilizing 5′ and 3′ modifications have been removed from the mRNA, the exposed RNA is degraded 5′-3′ by XRN exoribonucleases, and 3′-5′ by the exosome in all eukaryotes[19]. In Arabidopsis, dpc1, dcp2, vcs, xrn, and exosome mutations that compromise RQC allow aberrant RNA to accumulate and result in spontaneous RDR6-dependent PTGS of both transgenes[15,16,20–24] and endogenous genes[21,25].

Mutants impaired in the histone H3K4me3 demethylase JMJ14 have been identified in two independent forward genetic screens for PTGS-deficient mutants[26,27]. Molecular analysis of jmj14 plants compared to wild-type (WT) plants revealed a decrease in transgene siRNA accumulation, a decrease in transgene H3K4m3 and an increase in CHG methylation at the transgene promoter[26,28]. Thusfar, DNA methylation has only been mechanistically implicated in TGS of promoters in most eukaryotic lineages[7], inhibition of transcriptional elongation in fungi[29], and most recently, inhibition of aberrant transcription from the gene body, i.e., the intragenic regions, of plants[30,31] and mammalian embryonic stem cells[32]. Therefore, JMJ14 could link DNA methylation to aberrant RNAs and PTGS. However, the increase of transgene DNA methylation in jmj14 mutants contrasted the decreased methylation of CHH and CHG sites observed in two endogenous transposons in jmj14 mutants[27], and resembled more the effect of ibm1 mutations on endogenous sequences. IBM1 encodes a histone H3K9me2 demethylase, and ibm1 mutants have been reported to cause increased CHG and CHH methylation in the gene body of endogenous genes[33–35]. Therefore, the effect of jmj14 and ibm1 mutations on the epigenetic features of transgenic versus endogenous sequences required further investigation.

In this work, we perform genome-wide DNA methylation analysis of a jmj14 mutant and show that very few endogenous loci exhibit a change in DNA methylation in the mutant compared to wild type, and the DNA methylation changes in the jmj14 mutant are restricted to genomic sequences that normally exhibit highly variable DNA methylation levels[36,37]. In contrast, we detect increased methylation at all sequence contexts (CG, CHG, CHH) in transgenic sequences of the jmj14 mutant, but not the ibm1 mutant. We also show that JMJ14 promotes and IBM1 prevents transgene PTGS, respectively, and that the susceptibility of transgenes to PTGS correlates with low levels of DNA methylation in the transgene and high levels of transgene aberrant RNA, indicating that JMJ14 in combination with other chromatin modifying proteins plays a crucial role in establishing and maintaining stably expressed transgenes in an epigenetic state that is distinct from endogenous genes.

## Results

**A genetic screen for impaired systemic PTGS identifies additional jmj14 alleles.** Using the transgenic line 10027-3[38], a genetic screen recovered three jmj14 mutants, #38, #90 and #148, that showed defects in root-to-shoot PTGS transmission (Fig. 1A). The 10027-3 GFP reporter system involves a constitutive p35S: GFP transgene linked to another transgene that drives root tip-specific expression of a GF hairpin RNA homologous to GFP[38]. In 10027-3 WT plants, PTGS of GFP is initiated in the root apex during embryogenesis, and then as the seedlings germinate, silencing spreads into the shoot apex such that all true leaves that form show complete PTGS of GFP[38] (Fig. 1A). The systemic spreading of PTGS in this GFP reporter system requires RDR6-dependent amplification of dsRNA, using the constitutive GFP mRNA target as a template[38]. The three 10027-3 jmj14 mutants were identified by candidate gene sequencing on a collection of EMS-induced 10027-3 mutants showing defects in systemic PTGS[39]. Mutants #38 and #90 carry a nonsense and splice site mutation in JMJ14, respectively, and both mutants showed a lack of GFP silencing in the shoot apex but as the seedlings develop, PTGS of GFP spreads into the petiole and mid-vein of mature leaves (Fig. 1A). In contrast, mutant #148 contains a missense mutation in JMJ14 and showed a complete loss of GFP silencing (Fig. 1A).

F1 plants from crosses between mutants #38 and #90 (Fig. 1B), and between mutants #148 and #38 (Fig. 1C) displayed a defect in systemic PTGS, thereby confirming that the jmj14 mutations were the causative genetic defect in these mutants. These three mutants further substantiated the role of JMJ14 in RDR6-dependent PTGS[26], and suggest the possible involvement of JMJ14 in systemic spreading of PTGS. Hereafter, the #38, #90 and #148 mutants will be referred to as jmj14-5, jmj14-6 and jmj14-7, respectively (Fig. 1D).

**JMJ14 promotes aberrant RNA-based RDR6-dependent PTGS.** The results presented above suggest that JMJ14 plays a role in the systemic spreading of PTGS. However, they do not exclude that

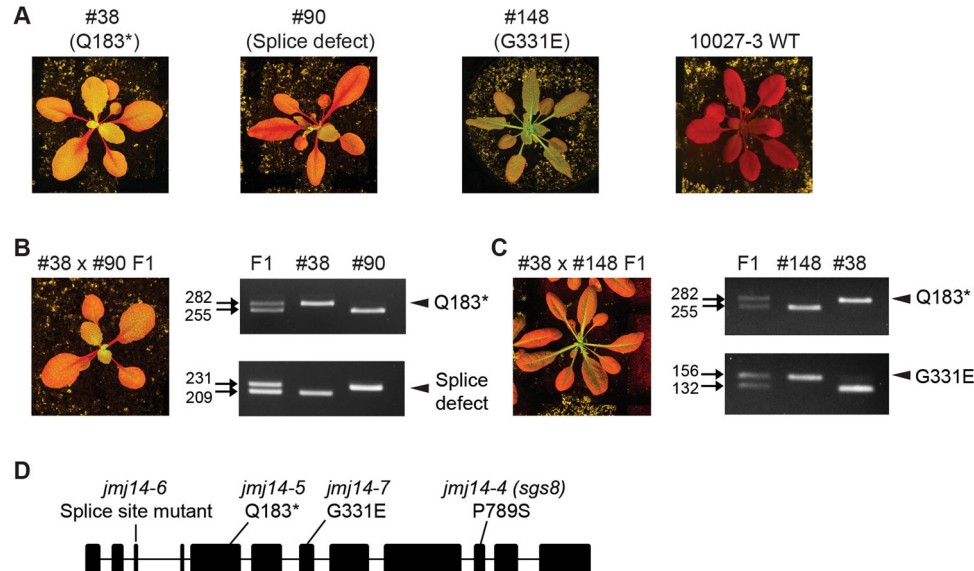

**Fig. 1 Three new alleles of *jmj14* show defects in systemic PTGS of *GFP*. A** Rosette GFP phenotypes of EMS#38 (Q183*; *jmj14-5*), EMS#90 (splicing defect; *jmj14-6*) and EMS#148 (G331E; *jmj14-7*). Transgenic line *10027-3* wild type (WT), the parent used in the genetic screen to identify *jmj14-5*, *jmj14-6* and *jmj14-7*, shows systemic PTGS of *GFP*. **B, C** EMS#90 (splice site mutant) and EMS#148 (G331E), respectively, were not complemented by EMS#38 (Q183*) as F1 plants from these crosses showed defective systemic PTGS. *JMJ14* PCR genotyping assays are shown in the right-hand side of panel **B**, **C**; arrows indicate the size in bps of the larger *jmj14* mutant allele and smaller WT allele for each PCR test and arrowheads indicate the diagnostic mutant *jmj14* allele compared to respective WT allele. The results for one F1 plant from each cross are shown in the Figure, however, we confirmed the phenotype and genotype of at least four F1 plants from each cross. **D** Location of the new *jmj14* mutations and the previously reported *jmj14* allele, *jmj14-4* (P789S)[26] in the *JMJ14* locus (AT4G20400). Exon and intron sequences are indicated by thick and narrow lines, respectively. Rosette images are of plants grown in soil under long-days for 4 weeks after planting. Uncropped and unprocessed scans of images of Fig. 1A–C are provided as a Source Data file.

JMJ14 also plays a role in the execution of PTGS. To resolve these two possibilities, the two-component GUS-silencing system *6b4-306* was used. This transgenic line consists of the *p35S:GUS* transgene locus called *6b4*, which never triggers PTGS alone in WT plants, plus an unlinked *p35S:hpGU* hairpin transgene locus called *306*, which produces dsRNA and siRNAs homologous to the first half of the *GUS* coding sequence[40]. In contrast to the *10027-3* GFP reporter system, both the *6b4* GUS transgene and *306* the hairpin transgene are expressed constitutively, resulting in the execution of PTGS in every cell, and therefore does not require RDR6-dependent spreading of PTGS to silence the *GUS* transgene throughout the plant[40]. Crossing the *6b4-306* double transgenic line to the *jmj14-4* mutant (P789S; Fig. 1D) produced triple homozygous *6b4-306 jmj14* plants in which PTGS occurred as efficiently as in *6b4-306* WT controls (Fig. 2A). Thus, JMJ14 is specifically required for RDR6-dependent PTGS, and not for the execution of PTGS when siRNAs are derived from a constitutively expressed hairpin transgene.

We next investigated the possibility that JMJ14 plays a role in the initiation of RDR6-dependent PTGS by enhancing the production of aberrant RNA from transgene loci. To address this question, we used the *6b4* transgenic line that does not carry any additional T-DNA loci. As mentioned above, the *6b4* locus carries a *p35S:GUS* transgene that never shows spontaneous PTGS in WT plants, but the introduction of *6b4* in various RQC-deficient mutant backgrounds results in the spontaneous triggering of PTGS[16,21,22,24]. This result is explained by the *6b4* GUS transgene producing low amounts of aberrant RNAs that are efficiently degraded by the cellular RQC pathways in WT plants. However, when RQC is impaired, aberrant RNAs derived from the *6b4* GUS transgene are converted into dsRNA by RDR6 and processed into 21- and 22 nt siRNAs by DCL4 and DCL2, respectively, thus activating PTGS. To determine if JMJ14 was contributing to the production of aberrant RNA from the *6b4*

*GUS* transgene, we utilized the mRNA decapping-defective *vcs-9* mutant to generate a *6b4 vcs jmj14* double mutant, which was compared to *6b4* WT, *6b4 jmj14* and *6b4 vcs* plants for the level of PTGS. The hypomorphic *vcs-9* allele was used because *vcs-9* mutants are viable and fertile, and because spontaneous PTGS occurs with 100% efficiency in *6b4 vcs-9* plants[21]. We observed an increase in GUS activity and a reduction in *GUS* siRNA accumulation in *6b4 vcs jmj14* compared to *6b4 vcs* plants (Fig. 2B). Given that JMJ14 is not involved in the production of siRNA derived from constitutively expressed hairpin RNA, these results strongly suggest that JMJ14 promotes PTGS by enhancing the production of aberrant RNA from the *6b4* GUS transgene.

**JMJ14 impairment decreases the production of *GUS* aberrant RNAs.** We previously identified an uncapped RNA, antisense to the *GUS* mRNA, hereafter referred to as aberrant *SUG* (*abSUG*) RNA[11]. Although it is not certain that this *abSUG* RNA is the aberrant RNA that triggers PTGS, its abundance correlates perfectly with the efficiency of PTGS in the two *p35S:GUS* reference lines *6b4* and *L1*. Indeed, it is detected at very low levels in *6b4* plants, which do not trigger PTGS spontaneously. It is also detected at only low levels in *6b4 xrn3 xrn4* and *L1* plants, which trigger PTGS spontaneously with 100% efficiency, most likely because the *abSUG* RNA is degraded by PTGS, similar to *GUS* mRNA. However, *abSUG* RNA accumulates to higher levels in *6b4 xrn3 xrn4 rdr6* and *L1 rdr6* plants, but remains low in *6b4 rdr6* plants. Given that the *p35S:GUS* transgene at the *L1* locus is transcribed at a higher level than its identical transgene counterpart at the *6b4* locus, these results suggest that the production of *abSUG* at a high level in *L1* plants or the absence of its degradation in *6b4 xrn3 xrn4* plants could explain the capacity of these plants to trigger PTGS[11].

To further test the hypothesis that *jmj14* mutations limits the production of aberrant RNAs, the abundance of *abSUG* RNA was

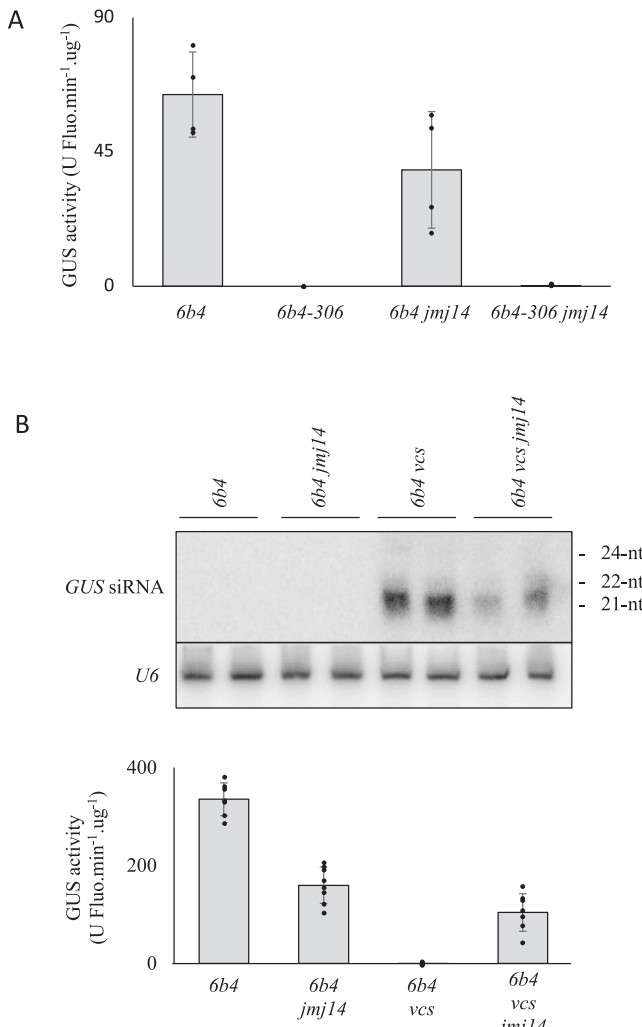

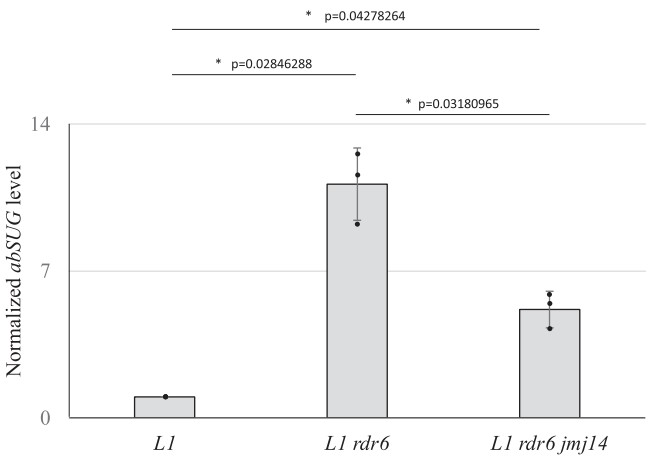

**Fig. 3 Impairing JMJ14 decreases the steady state level of aberrant *abSUG* RNA.** Normalized expression of *abSUG* RNA in *L1, L1 rdr6* and *L1 rdr6 jmj14* plants. Results are expressed as a fold change compared to *L1* (*L1* = 1) and normalized to *eIF1a*. Error bars represent the standard deviation of three replicates. The significance was assayed with a Bonferroni corrected *t*-test (*P* < 0.05).

hypothesis that JMJ14 promotes transgene-derived aberrant RNA production.

**A synergistic relationship between systemic PTGS signals and aberrant RNA in the induction of RDR6-dependent PTGS.** The results described above strongly suggest that JMJ14 promotes the production of aberrant RNA from transgene loci, and that JMJ14 also plays a role in systemic PTGS. To investigate the possibility of a collaborative relationship between aberrant RNA produced from transgene loci and systemic PTGS signals in the induction of RDR6-dependent PTGS, we conducted reciprocal grafting experiments using WT and *jmj14* mutants as rootstocks and scions in our GFP and GUS reporter systems for graft-transmissible PTGS[13,38].

When GFP-expressing *10027-3 jmj14-7* scions were grafted onto GFP-silenced *10027-3* WT rootstocks, graft-transmissible PTGS failed to be initiated (Fig. 4A), suggesting that the *jmj14-7* mutation completely abolished the capacity of the scion to respond to the systemic PTGS signal transmitted from the *10027-3* WT roots. A feature of PTGS in plants is the predominance of 21 nt siRNAs produced by DCL4, along with a much lower abundance of 22 nt siRNA produced by DCL2. Nevertheless, DCL2 and its 22 nt siRNA play a more important role in systemic PTGS than DCL4[38,41]. Therefore, we also grafted *10027-3 jmj14-7* scions onto *10027-3 dcl4-5* rootstocks that produce predominantly DCL2-dependent 22 nt siRNAs[38], and again, graft-transmissible PTGS failed to be initiated in the *jmj14-7* scions. These results suggest that *jmj14* scions are incapable of responding to either DCL4-dependent 21 nt or DCL2-dependent 22 nt siRNAs as mobile silencing signals.

We also grafted GFP-expressing scions of transgenic line *214* (Supplementary Fig. 1) onto *10027-3* or *10027-3 jmj14* mutant lines as rootstocks[13]. Whereas *GFP* PTGS is efficiently transmitted from *10027-3* rootstocks to *214* scions, we observed a decrease in systemic transmission of PTGS when *214* scions were grafted onto *10027-3 jmj14* rootstocks (Fig. 4A).

These findings were confirmed using the *jmj14-4* mutant allele in the *6b4* GUS reporter background. Whereas grafting *6b4* WT scions grafted onto *L1* WT rootstocks triggered efficient *GUS* PTGS in *6b4* scions, *6b4 jmj14-4* scions grafted onto *L1* WT rootstocks did not trigger PTGS (Fig. 4B, C). In addition, *6b4 jmj14-4* scions also failed to initiate PTGS when grafted onto

**Fig. 2 JMJ14 promotes aberrant RNA-induced PTGS, but is not required for PTGS induced by hairpin dsRNA. A** Effect of JMJ14 impairment on the transgenic *6b4* GUS-expressing line and the *6b4-306* GUS-silenced line in which *GUS* mRNA produced by the *6b4* locus is silenced by the consitutive expression of *GUS* hairpin dsRNA by the *306* locus. No PTGS is observed in *6b4* and *6b4 jmj14* plants, whereas PTGS occurs efficiently in both *6b4-306* and *6b4-306 jmj14* plants. Results represent the mean of four replicates. Errors bars represent the standard deviation of these four replicates. **B** Effect of JMJ14 impairment on the transgenic *6b4* GUS-expressing line and the *6b4 vcs* GUS-silenced line in which the *6b4 GUS* transgene undergoes aberrant RNA-induced PTGS due to the loss of RQC. PTGS occurs efficiently in *6b4 vcs* plants but is reduced in *6b4 vcs jmj14* plants, revealed by reduced accumulation of *GUS* siRNAs and increased GUS activity in *6b4 vcs jmj14* compared to *6b4 vcs* plants. Results represent the mean of eight replicates. Errors bars represent the standard deviation of these eight replicates. Uncropped and unprocessed scans of images in Fig. 2B are provided as a Source Data file.

measured in *L1, L1 rdr6* and *L1 jmj14 rdr6* plants. To reliably assess the effect of JMJ14 on the levels of the *abSUG* RNA, the *rdr6* mutant background was used to avoid RDR6-dependent PTGS of the *abSUG* RNA. The amount of *abSUG* RNA was very low in *L1* WT plants compared to *L1 rdr6* plants, thereby confirming that *abSUG* RNA is degraded by PTGS in *L1* WT plants (Fig. 3). However, the amount of *abSUG* RNA in the *L1 rdr6* mutant was at least double the amount in the *L1 jmj14 rdr6* double mutant (Fig. 3), which further supports the

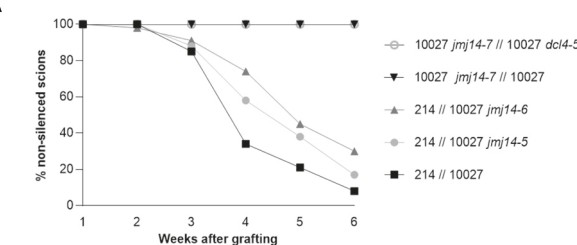

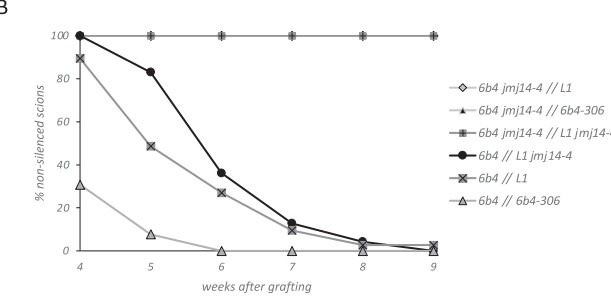

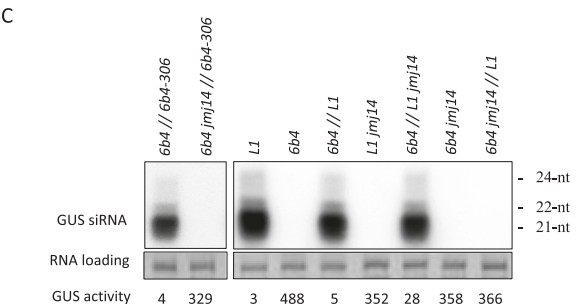

**Fig. 4 JMJ14 is required for reception of graft-transmissible PTGS in scions and efficient transmission of PTGS from rootstocks. A** Grafting results using the transgenic *214* GFP-expressing and *10027-3* GFP-silencing system for systemic PTGS. When *GFP*-expressing *10027-3 jmj14-7* scions were grafted onto either *10027-3* wild-type (WT) rootstocks that produce primarily DCL4-dependent 21 nt siRNAs or *10027-3 dcl4-5* rootstocks producing predominantly DCL2-dependent 22 nt siRNAs[38], no systemic PTGS was observed. Similarly, when this most severe *jmj14* allele, *jmj14-7*, was used as rootstocks and grafted to *GFP*-expressing *214* WT scions, no transmission of systemic PTGS was observed. Weaker alleles of *jmj14*, i.e., *jmj14-5* and *jmj14-6*, were also used as rootstocks grafted onto *GFP*-expressing *214* WT scions[38], and these mutant rootstocks showed delayed transmission of systemic PTGS compared to WT rootstocks. In total, 22 to 66 grafted plants were assessed over at least two independent experiments for each combination of grafted genotypes. **B** Grafting results using the transgenic *GUS*-expressing *6b4* line and *GUS*-silencing *6b4-306* and *L1* lines as a system for systemic PTGS. When *6b4* scions were grafted onto *6b4-306*, *L1* or *L1 jmj14-4* rootstocks, systemic PTGS was observed, although delayed when using *L1 jmj14-4* rootstocks. In contrast, when *6b4 jmj14-4* scions were grafted onto *6b4-306* or *L1* rootstocks, no systemic PTGS was observed. In total, 16 to 74 grafted plants were assessed over at least two independent experiments for each combination of grafted genotypes. **C** GUS activity (units/μg protein/min) and *GUS* siRNA accumulation in ungrafted *6b4*, *6b4 jmj14*, *L1*, *L1 jmj14* lines and *6b4 // 6b4-306*, *6b4 jmj14 // 6b4-306*, *6b4 // L1*, *6b4 // L1 jmj14* and *6b4 jmj14 // L1* grafts. Each sample corresponds to a mix of four plants. The two blots were hybridized with the same probe and exposed the same amount of time. Uncropped and unprocessed scans of images in Fig. 4C are provided as a Source Data file.

*6b4-306* rootstocks (Fig. 4C). Furthermore, similar to what was observed for *jmj14-5* and *jmj14-6* rootstocks using GFP as the reporter for PTGS (Fig. 4A), when GUS-expressing *6b4* WT scions were grafted onto *L1 jmj14-4* rootstocks, the onset of graft-transmissible PTGS in the scions was delayed (Fig. 4B). Altogether, these results indicate that JMJ14 plays an essential role in the perception of the PTGS systemic signal.

**Transgene DNA methylation counteracts aberrant RNA-induced PTGS.** Southern blot analysis using a methylation-sensitive enzyme previously revealed that *L1 jmj14-3*, *L1 jmj14-4*, *6b4 jmj14-3*, and *6b4 jmj14-4* plants exhibit increased DNA methylation at CHG sites in the promoter of the *p35S:GUS* transgene[26]. To determine the methylation profile of the entire transgene locus in a *jmj14* mutant background, whole-genome bisulfite sequencing (WGBS) analysis was performed on plants carrying the *6b4* locus. WGBS revealed that the transgenic line *6b4* carried a single T-DNA insertion. However, the left border did not delineate the *6b4* T-DNA insertion and the entire binary vector was also found to have inserted adjacent to the left border (Supplementary Fig. 2). This phenomenon of co-transfer of non-T-DNA vector sequences occurs frequently with T-DNA integration events into plant chromosomes[42]. WGBS analysis revealed a high level of CG methylation in the *GUS* coding sequence of *6b4* plants (Fig. 5), which most likely reflects the highly transcribed status of the *p35S:GUS* transgene in this transgenic line. Of note, almost no CG methylation was found in the adjacent *NPTII* transgene (Supplementary Fig. 3), consistent with a low level of *NPTII* expression and the *6b4* line being barely resistant to kanamycin[11]. However, a surprisingly high level of CG methylation was observed in the non-T-DNA vector sequences inserted along with the T-DNA (Supplementary Fig. 3), suggesting that these sequences had been transcribed by a plant RNA polymerase prior to, and/or after integration into the chromosome. In contrast to CG methylation, the level of CHH and CHG methylation was very low in both T-DNA and vector sequences of the *6b4* locus in the WT genetic background (Fig. 5 and Supplementary Fig. 3). The situation was very different in *6b4 jmj14-4* plants where the *GUS* coding sequence exhibited CHG hypermethylation and the *35S* promoter showed increased CHG and CHH methylation (Fig. 5). Of note, CHH and CHG hypermethylation in *6b4 jmj14* was not observed in the adjacent *NPTII* transgene but was observed in the vector sequences flanking the left border (Supplementary Fig. 3A), suggesting that CHH and CHG hypermethylation in *jmj14* occurs mostly at sequences that show high levels of CG methylation in the WT genetic background.

These results prompted us to hypothesize that CHH and CHG hypermethylation of the *p35S:GUS* transgene in *6b4 jmj14* plants could prevent, or at least reduce, the production of aberrant RNAs (Fig. 3), thus explaining why *jmj14* scions cannot trigger PTGS upon grafting onto silenced rootstocks (Fig. 4). To test this hypothesis, *6b4 jmj14 drm2 cmt3* plants were generated, and we confirmed the absence of CHH and CHG methylation in the *p35S:GUS* transgene and the entire *6b4* locus in this line (Fig. 5 and Supplementary Fig. 3). When grafted onto *GUS*-silenced rootstocks, *6b4 jmj14 drm2 cmt3* plants triggered PTGS as efficiently as grafted *6b4* WT controls (Table 1), supporting the hypothesis that CHH and CHG hypermethylation is instrumental in preventing systemic PTGS in *6b4 jmj14* plants.

There is a low but detectable level of CHG methylation (but not CHH methylation) in the *GUS* coding sequence of *6b4* WT plants (Fig. 5), and we asked if this residual level of CHG methylation prevents spontaneous induction of PTGS in these

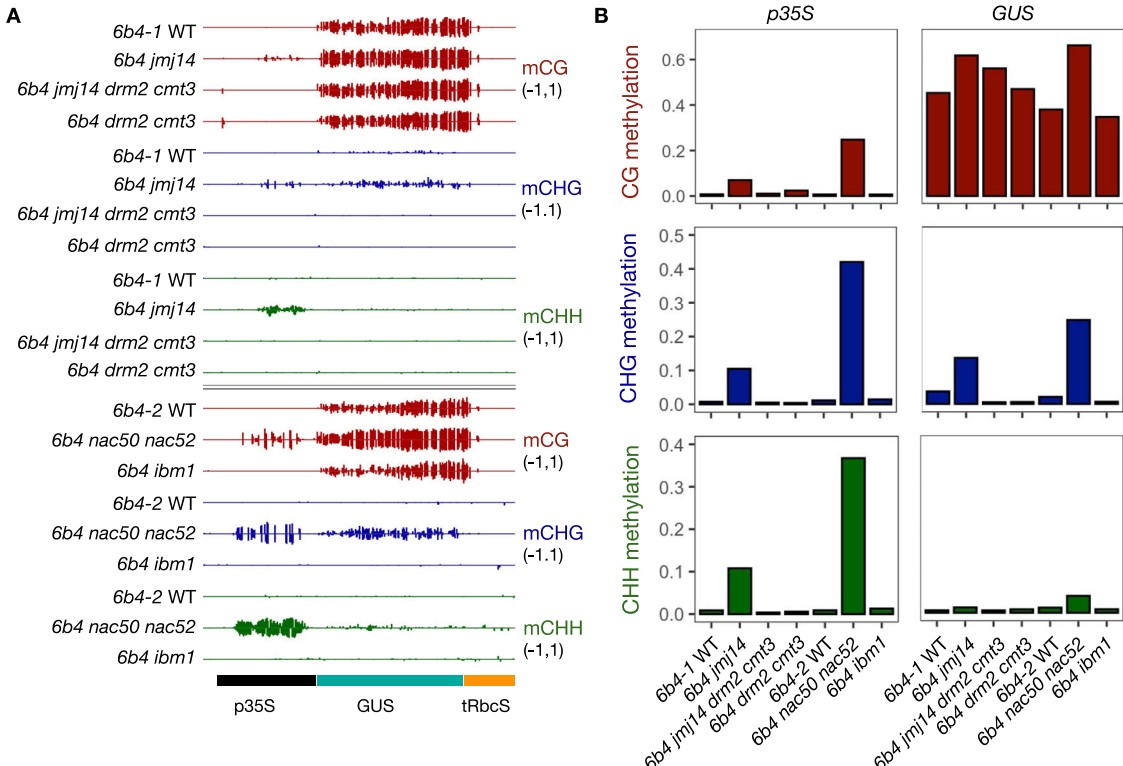

**Fig. 5 JMJ14 and NAC50/NAC52 prevent CHG and CHH methylation of the *p35S:GUS* transgene. A** CG, CHG and CHH methylation (where H = A, C or T) profiles for the *6b4 GUS* transgene consisting of a modified *35S* promoter (*p35S*), the *GUS* coding sequence (*GUS*) and a *Rubisco* terminator (*tRbcS*). The top panel shows DNA methylation profiles of 5-week-old plants for *6b4* WT control 1 (*6b4-1* WT), and mutants *6b4 jmj14*, *6b4 jmj14 drm2 cmt3* and *6b4 drm2 cmt3*. The bottom panel shows DNA methylation profiles of 8-week-old plants for *6b4* WT control 2 (*6b4-2* WT) and mutants *6b4 nac50 nac52* and *6b4 ibm1*. Bisulfite data were analyzed using R version 4.0.2. **B** Proportion of methylated cytosines in the three sequence contexts across the *p35S* and *GUS* sequences show that increased CHG and CHH methylation in *6b4 jmj14* and *6b4 nac50 nac52* plants is dependent on the DRM2-CMT3 pathway, and that impairment of IBM1, if anything, decreases rather than increases DNA methylation of the *GUS* coding sequence.

---

**Table 1 Frequency of spontaneous and graft-induced PTGS.**

| Genotype | Long-day | % spontaneous PTGS | Short-day | % spontaneous PTGS | % graft-induced PTGS |
|---|---|---|---|---|---|
| *6b4 nac50 nac52* | BR1 | 0 (*n* = 48) | BR1 | 0 (*n* = 6) | 0 (*n* = 24) |
| | BR2 | 0 (*n* = 48) | BR2 | 0 (*n* = 6) | 0 (*n* = 24) |
| *6b4 jmj14* | BR1 | 0 (*n* = 48) | BR1 | 0 (*n* = 16) | 0 (*n* = 13) |
| | BR2 | 0 (*n* = 48) | BR2 | 0 (*n* = 16) | 0 (*n* = 27) |
| *6b4* WT | BR1 | 0 (*n* = 48) | BR1 | 0 (*n* = 39) | 100 (*n* = 36) |
| | BR2 | 0 (*n* = 48) | BR2 | 0 (*n* = 32) | 100 (*n* = 48) |
| *6b4 jmj14 cmt3 drm2* | BR1 | 0 (*n* = 48) | BR1 | 0 (*n* = 26) | 96 (*n* = 26) |
| | BR2 | 0 (*n* = 48) | BR2 | 0 (*n* = 30) | 100 (*n* = 26) |
| *6b4 cmt3 drm2* | BR1 | 12 (*n* = 48) | BR1 | 9 (*n* = 11) | 100 (*n* = 8) |
| | BR2 | 8 (*n* = 48) | BR2 | 12 (*n* = 8) | 100 (*n* = 8) |
| *6b4 ibm1* | BR1 | 19 (*n* = 48) | BR1 | nd | nd |
| | BR2 | 23 (*n* = 48) | BR2 | nd | nd |

Note: For each genotype tested, the frequency of spontaneous PTGS was first determined in long-day conditions. Two independent experiments were performed, each consisting of 48 plants. For grafting experiments, plants were grown in short-day conditions. Two independent experiments were performed for each grafting combination. In each experiment, non-grafted plants were kept as controls and analyzed to determine the frequency of spontaneous PTGS in short-day conditions. Plants were considered silenced when exhibiting <20 fluorescent GUS activity units per minute per microgram of total protein.
*BR* biological replicate, *nd* not determined.

---

plants. To test this hypothesis, *6b4 drm2 cmt3* plants were generated. Our analysis revealed that CHG methylation is reduced in the *GUS* coding sequence of *6b4 drm2 cmt3* plants compared to *6b4* WT plants (Fig. 5A), and indeed, ~10% of these plants trigger spontaneous PTGS (Table 1). These results strongly support the hypothesis that transgene CHG methylation strongly limits the production of aberrant RNAs that would otherwise induce spontaneous PTGS of the homologous *GUS* coding

sequence. In *6b4 drm2 cmt3* plants, we propose that the lower level of CHG methylation of the *GUS* coding sequence increases the transcription of aberrant RNAs, thereby allowing spontaneous PTGS to occur. While *6b4* WT plants produce insufficient aberrant RNAs to trigger spontaneous PTGS of the *GUS* coding sequence, it is sufficient for graft-induced PTGS to occur due to a synergistic relationship between aberrant RNA expressed from the *GUS* transgene of *6b4* WT scions and systemic PTGS signals

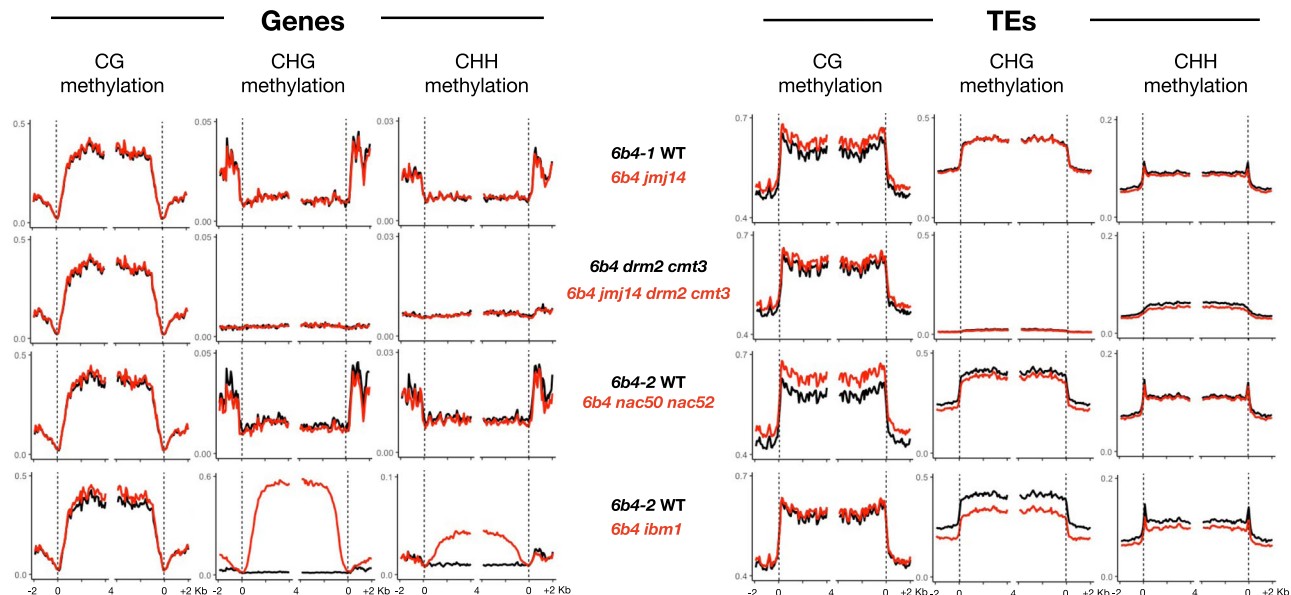

**Fig. 6 JMJ14 and NAC50/NAC52 do not affect CHG and CHH methylation of endogenous genes and TEs.** The gene body of endogenous genes (left panels) and TEs (right panels) were annotated according to the TAIR10 reference genome and aligned at the 5′ and 3′ ends (dashed lines), and average CG, CHG and CHH methylation levels for 100-bp intervals were plotted in pairwise comparisons between *6b4-1* WT and *6b4 jmj14*, and *6b4 drm2 cmt3* and *6b4 jmj14 drm2 cmt3* of 5-week-old plants, and *6b4-2* WT and *6b4 nac50 nac52*, and *6b4-2* WT and *6b4 ibm1* for 8-week-old plants. This analysis showed that CHG and CHH methylation in the body of endogenous genes and TEs is not affected in *6b4 jmj14* and *6b4 nac50 nac52* plants, whereas CHG and CHH is substantially increased in the body of endogenous genes in *6b4 ibm1* plants. Bisulfite data were analyzed using R version 4.0.2.

derived from *GUS*-silenced rootstocks. In contrast, in *6b4 jmj14* scions, increased CHG methylation further limits the production of aberrant RNA from the *6b4 GUS* transgene, rendering it non-responsive to graft-transmissible PTGS signals from silenced *GUS* rootstocks.

Remarkably, WGBS of *6b4* WT and *6b4 jmj14-4* plants revealed that loss of JMJ14 had a limited effect on the extent of DNA methylation at endogenous loci (Fig. 6). A summary of all endogenous DMRs identified between *6b4* WT and *6b4 jmj14-4* is presented in Supplementary Data 1. Only one endogenous hyper CHG DMR was identified in *jmj14*, and this endogenous region did not show hyper CG and hyper CHH methylation, whereas the transgenic sequences (both the T-DNA and the co-integrated backbone sequences of the binary vector) exhibited hyper CG, hyper CHG and hyper CHH methylation (Fig. 5 and Supplementary Fig. 3).

**Both JMJ14 and NAC52 interact with the *p35S:GUS* transgene and are required for systemic PTGS.** Mutations in *JMJ14* were previously shown to adversely affect Pol II occupancy and H3K4me3 levels in the promoter of the *p35S:GUS* transgene in transgenic line *6b4*[26]. However, it was not determined if this effect was due to a direct interaction between JMJ14 and chromatin at the *35S* promoter. To address this question, chromatin immunoprecipitation (ChIP) of JMJ14 was performed on the *6b4 jmj14-4* line complemented with a *pJMJ14:3xFlag-JMJ14* transgene[43]. ChIP was conducted using Flag antibodies and followed by quantitative PCR (qPCR) using primer pairs located in the *35S* promoter, and in the 5′ and 3′ regions within the *GUS* transgene body. Quantitative ChIP-PCR revealed a significant enrichment of JMJ14 primarily at the *35S* promoter (Fig. 7A). These results indicate that JMJ14 is a component of chromatin at the promoter of the *p35S:GUS* transgene, and are consistent with Pol II occupancy, high levels of H3K4me3[26] and lack of DNA methylation (Fig. 5) within the *35S* promoter of *6b4* WT plants.

JMJ14 has also been shown to interact with the NAC-domain transcription factor NAC52 and its close relative NAC50[44,45]. NAC50 and NAC52 most likely play redundant roles, but NAC52 appears more important than NAC50 in *Arabidopsis* due to its higher level of expression. Indeed, a genetic screen for PTGS impaired mutants identified a mutation, originally named *sgs1*, which turned out to be a mutant allele of *NAC52*, whereas reverse genetics showed that a *nac50* knockout mutation had no effect on PTGS[28]. To determine whether *NAC50* or *NAC52* participates in graft-induced PTGS of the *6b4 GUS* transgene, a *6b4 nac50 nac52* double mutant was generated and grafted onto *GUS*-silenced rootstocks. Similar to *6b4 jmj14*, the *6b4 nac50 nac52* scions were incapable of triggering PTGS of *GUS* when grafted onto *GUS*-silenced rootstocks (Table 1), indicating that both *JMJ14* and *NAC50/NAC52* are required for reception of graft-induced PTGS of the *6b4 GUS* transgene. In addition, WGBS analysis of non-grafted *6b4 nac50 nac52* plants revealed transgene CHH and CHG hypermethylation similar to non-grafted *6b4 jmj14* plants (Fig. 5B and Supplementary Fig. 3B), confirming the correlation between CHH and CHG hypermethylation and decreased induction of PTGS.

Importantly, WGBS of *6b4* WT and *6b4 nac50 nac52* plants also confirmed that like the loss of JMJ14, loss of both NAC50 and NAC52 had a limited effect on the extent of DNA methylation at endogenous loci. Indeed, only one endogenous hyper CHG DMR was identified in *jmj14* and only one endogenous hyper CHG DMR was identified in *nac50 nac52* (Supplementary Data 1). These two DMRs were located on chromosome 5 and chromosome 3, respectively, indicating that *jmj14-* and *nac50 nac52*-dependent hyper CHG methylation is only found in the transgenic sequences (both the T-DNA and the co-integrated backbone sequences of the binary vector). Moreover, transgenic sequences not only exhibited hyper CHG methylation but also hyper CG and hyper CHH methylation (Fig. 5 and Supplementary Fig. 3). The only endogenous hyper CHG methylation found in *jmj14* did not show hyper CG and

A

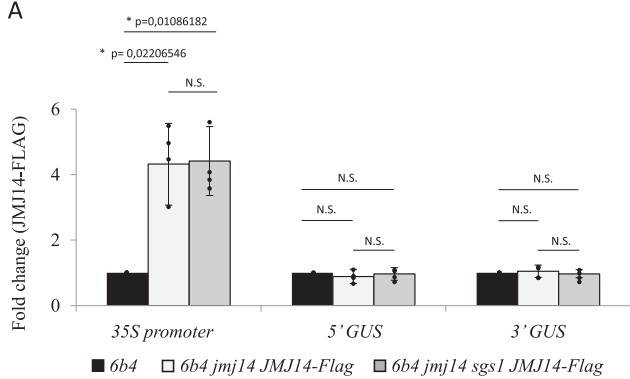

B

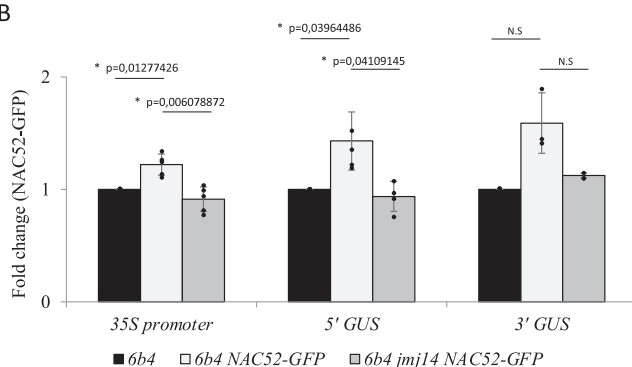

**Fig. 7 Hierarchical binding of JMJ14 and NAC52 to the promoter and coding sequence of the *p35S:GUS* transgene, respectively. A** ChIP-qPCR analyses were performed on the aerial part of 17-day-old seedlings of the indicated genotypes using a Flag antibody to immunoprecipitate JMJ14-Flag. Levels are given as percentages of IP$^{sample}$/IP$^{6b4}$ relative to GAPDH, which was used as a control. The graphical representation shows the fold change as the mean of four replicates. Error bars represent the standard deviation of these four replicates. The significance was assayed with a Bonferroni corrected two-sides *t*-test (*P* < 0.05). Results indicate a relative enrichment of JMJ14 on the *35S* promoter of the *6b4* locus, which was not affected by the *sgs1* mutation impairing NAC52. **B** ChIP-qPCR analyses were performed on the aerial part of 17-day-old seedlings of the indicated genotypes using a GFP antibody to immunoprecipitate NAC52-GFP. Levels are given as percentages of IP$^{sample}$/IP$^{6b4}$ relative to GAPDH, which was used as a control. The graphical representation shows the fold change as the mean of five replicates. Error bars represent the standard deviation of these five replicates. The significance was assayed with a Bonferroni corrected two-sides *t*-test (*P* < 0,05). Results indicate a relative enrichment of NAC52 on the *35S* promoter and the beginning of the *GUS* coding sequence of the *6b4* locus, which was abolished by the *jmj14* mutation, indicating a hierarchical action of JMJ14 and NAC52 at the *6b4* locus.

hyper CHH methylation, while the only endogenous hyper CHG methylation found in *nac50 nac52* showed hyper CHH methylation but not hyper CG methylation (Supplementary Data 1). Therefore, it is most likely that the endogenous DMRs observed between wild type and either *jmj14* or *nac50 nac52* correspond to endogenous genomic sequences that exhibit highly variable methylation levels[36,37]. Together, WGBS analysis of WT, *jmj14* and *nac50 nac52* plants indicates that JMJ14 and NAC50/NAC52 specifically limit the level of DNA methylation of transgenes and have essentially no effect on the DNA methylation status of endogenous genes.

To further address the role of NAC50/NAC52 in PTGS, a *pUBQ10:NAC52-GFP* transgene was generated and introduced into the *6b4* WT line. The *pUBQ10* promoter drives constitutive

expression of transgenes in *Arabidopsis*. At first, we tested whether the *pUBQ10:NAC52-GFP* constructs expressed a functional protein that actually interacts with JMJ. To do so, co-immunoprecipitation (co-IP) experiments were performed on plants resulting from a cross between a *pJMJ14:JMJ14-Myc* line[46] and the *6b4 + pUBQ10:NAC52-GFP* line. As expected, JMJ14-Myc and NAC52-GFP proteins co-precipitated in pull-down assays using a GFP antibody (Supplementary Fig. 4). Furthermore, ChIP was conducted on the *6b4 + pUBQ10:NAC52-GFP* line using GFP antibodies, followed by qPCR using primer pairs located in the *35S* promoter, and in the 5′ and 3′ regions within the *GUS* transgene body. The ChIP-qPCR analysis revealed a significant enrichment of NAC52 in the *35S* promoter and the 5′ region of the *GUS* coding sequence (Fig. 7B).

Finally, to address the hierarchical action of JMJ14 and NAC50/NAC52 in PTGS, we generated *6b4 jmj14 nac52 + pJMJ14:3xFlag-JMJ14*[43] and *6b4 jmj14 + pUBQ10:NAC52-GFP* lines. ChIP-qPCR revealed that the association of JMJ14 with chromatin at the *35S:GUS* transgene was not affected by the *nac52(sgs1)* mutation (Fig. 7A), whereas the association of NAC52 was impaired by the *jmj14* mutation (Fig. 7B). Altogether, these results strongly suggest that JMJ14 binds to chromatin at the promoter of the *p35S:GUS* transgene, which promotes the binding of NAC50/NAC52 to the *35S* promoter and the downstream *GUS* coding sequence. Our results are consistent with JMJ14 and NAC50/NAC52 acting in a chromatin complex to prevent CHG methylation throughout the entire transgene, thus allowing aberrant RNAs to be produced, which in turn induce PTGS.

**IBM1 impairment slightly decreases CHH and CHG methylation in the transgene body and promotes PTGS.** To test further the hypothesis that transgene methylation influences PTGS capacity, the *6b4* locus was introduced into the *ibm1* mutant background. *IBM1* encodes an H3K9me2 demethylase that suppresses CHH and CHG methylation in the body of about half of the endogenous genes of *Arabidopsis*[33–35,47]. We therefore expected *6b4 ibm1* plants to exhibit increased CHH and CHG methylation in the *GUS* coding sequence and suppress induction of PTGS when grafted onto silenced *GUS* rootstocks. However, while WGBS analysis confirmed increased CHH and CHG methylation in the body of 9000 endogenous genes (Fig. 6 and Supplementary Data 1), CHH and CHG methylation remained the same or slightly decreased in transgenic sequences in *6b4 ibm1* plants (Fig. 5 and Supplementary Fig. 3B). The decrease in CHG methylation in transgenic sequences of *6b4 ibm1* plants was most obvious in the *GUS* coding sequence (Fig. 5B). These results confirmed that actively expressed transgenes and endogenous genes exhibit major epigenetic differences. Moreover, spontaneous triggering of *GUS* PTGS occured in ~20% of *6b4 ibm1* plants (Table 1 and Supplementary Fig. 5), indicating that JMJ14 and IBM1 have opposite effects on transgene PTGS.

## Discussion

Endogenous protein-coding genes produce aberrant RNAs that are efficiently degraded by RNA quality control (RQC) pathways. In plants, this is essential to prevent the conversion of aberrant RNAs into dsRNA by RDRs, which would result in the production of siRNAs that could destroy homologous and essential mRNAs. Indeed, RQC impairment causes lethality in *Arabidopsis*, which can be rescued by suppressing RDR6 activity[21,25]. Similarly, transgenes produce aberrant RNAs in proportion to the level of transcript, and provided the aberrant RNAs do not exceed an abundance threshold within the cell, they are also eliminated by RQC. However, if RQC is impaired or saturated by high levels

of transcription, both transgenes and endogenous genes can be subjected to RDR6-dependent PTGS[15,16]. Occasionally, endogenous loci in plants can be subjected to PTGS despite a functioning RQC pathway being present. However, these loci generally exhibit genomic rearrangements involving gene duplication events that allow the production of dsRNA and induction of PTGS[48–54]. It is assumed that such rearrangements involving endogenous genes are tolerated because they only adversely affect dispensable genes. Obviously, transgenes are also dispensable because they correspond to sequences that do not exist naturally in plants, and the transgene loci that induce PTGS the most efficiently involve strong promoters and/or tandem insertion of multiple copies of the transgene, often in inverted orientations[55]. High rates of transcription from multiple transgenes increases the chances of aberrant RNA by-products forming, and only a fraction of this aberrant RNA needs to escape degradation by RQC to be recruited by RDR6, and converted into dsRNA to induce PTGS.

Despite these similarities, actively expressed transgenes are generally more susceptible to PTGS than endogenous genes. These differences between actively expressed transgenes and endogenous genes have long remained a mystery. Our finding that transgenes and endogenous genes exhibit different epigenetic characteristics is particularly interesting because transgenes are transferred into the host genome as naked DNA or naked DNA associated with *Agrobacterium* proteins, and once integrated into the chromosome, must associate with histones to form chromatin. At present, the type of eukaryotic histones that associate with naked or newly integrated foreign DNA is not known. One could imagine that the insertion of extrachromosomal DNA into the chromosome attracts particular histone marks to label this DNA as 'new or foreign'. It seems likely that what happens for transgenes also occurs when a new copy of a retrotransposon integrates into the genome. Supporting this hypothesis, it has been shown that activation of retrotransposons triggers a PTGS response, just like transgenes do[56–58]. Although this hypothesis is attractive, our findings suggest that there are striking epigenetic differences between transgenes and endogenous loci, including genes and stabilized transposons. Here, we showed that transgenes and endogenous loci (genes and stabilized transposons) behave differently when the histone H3K4me3 demethylase JMJ14, and two cooperating transcription factors that interact with JMJ14, NAC50, and NAC52, are impaired. Indeed, in *jmj14* or *nac50 nac52* mutants, an increase in CHG methylation, and CHH methylation to a lesser extent, is observed at the *6b4* transgene locus (Fig. 5 and Supplementary Fig. 3). An increase in CHG methylation was particularly evident in the promoter and gene body of the *p35S:GUS* transgene, and also parts of the integrated *Agrobacterium* vector sequences (Fig. 5 and Supplementary Fig. 3), whereas no change in DNA methylation profiles was observed at any endogenous genomic loci in these mutants (Fig. 6 and Supplementary Data 1). Remarkably, the *jmj14* and *nac50nac52* mutations also suppress the triggering of PTGS of the *p35S:GUS* and *p35S:GFP* when these transgenic mutant lines were used as scions grafted onto rootstocks showing PTGS (Fig. 4 and Table 1). Moreover, CHH and CHG hypermethylation, and not simply the absence of JMJ14 or NAC50/NAC52, appears causal in preventing PTGS because *6b4 jmj14 drm2 cmt3* scions trigger PTGS when grafted onto silenced rootstocks (Table 1), indicating that the impairment of DRM2-CMT3-dependent CHH and CHG methylation in a *jmj14* background restores the susceptibility of transgenes to graft-transmissible PTGS. As JMJ14 and NAC52 bind to the *p35S:GUS* transgene within the *6b4* locus (Fig. 7), we propose that the interacting complex limits CHG and CHH methylation of the *GUS* coding sequence, allowing the production of aberrant RNAs that are required to trigger and maintain PTGS.

Supporting this hypothesis, the *6b4 drm2 cmt3* transgenic line lacks CHH and CHG methylation of the *GUS* coding sequence and triggers PTGS spontaneously, whereas the *6b4* WT line exhibits a low CHH and CHG methylation level of the *GUS* coding sequence that is nevertheless sufficient to block the spontaneous triggering of PTGS.

The *6b4* T-DNA insertion locus contains not only an intact T-DNA but also the entire *Agrobacterium* binary vector sequence alongside the left border of the T-DNA (Supplementary Fig. 2). Remarkably, the CHG and CHH methylation profiles of the vector sequences were similar to the *p35S:GUS* transgene in WT plants and increased in the *jmj14* mutant in a DRM2-CMT3-dependent manner (Supplementary Fig. 3). These results suggest that not just promoter-driven transgenes, but also promoterless foreign DNA sequences could be subject to JMJ14-dependent suppression of CHG and CHH methylation, resulting in the production of aberrant RNA and potential PTGS of homologous mRNA. It is interesting to note that bombardment of transgenic tobacco expressing a *p35S:GFP* transgene with promoterless *GFP* DNA sequences can induce PTGS[59]. Adding to this earlier report, our data are consistent with promoter-independent aberrant RNA transcripts being produced from foreign DNA sequences prior to or immediately after integration into the plant chromosome. In contrast to the *p35S:GUS* transgene and *Agrobacterium* vector sequences, CHG and CHH methylation of the *pNOS:NPTII:tNOS* transgene within the *6b4* locus remained low in the *jmj14* and *nac50 nac52* mutant backgrounds (Supplementary Fig. 3). This could be explained by JMJ14 not being able to bind to the *Agrobacterium NOS* promoter, which may have been selected for during the evolution of *Agrobacterium* as a pathogen to ensure that the transferred oncogenes on the T-DNA were not epigenetically distinguishable from endogenous plant genes.

H3K4 trimethylation and DNA methylation are considered to have antagonistic influences activating and repressing expression of endogenous genes, respectively[46]. Therefore, it was expected that for endogenous loci in *jmj14* mutants, where H3K4me3 levels increase, DNA methylation would decrease. However, our WGBS analysis revealed that there were minimal changes in DNA methylation levels within endogenous genes of *jmj14* (Fig. 6), further emphasizing the difference between transgenes and endogenous sequences. Nevertheless, we asked whether the binding of JMJ14 could promote the production of aberrant RNAs from endogenous genes and their subsequent transformation into siRNAs, like it does at the *p35S:GUS* transgene of the *6b4* locus. If this was the case, one would expect that the fraction of endogenous genes that produce siRNAs when RQC is impaired[21,25,60] would be enriched for genes that bind JMJ14[45]. However, the fraction of siRNA-producing endogenous genes that bind JMJ14 is exactly that expected by chance (Supplementary Fig. 6), whether considering the siRNA-producing endogenous genes identified in the decapping mutants *dcp2* and *vcs*[21], the siRNA-producing endogenous genes identified in the exonuclease double mutant *xrn4 ski2*[25], or the siRNA-producing endogenous genes identified when plants are infected by viruses that inhibit RQC[60]. Therefore, unlike the *p35S:GUS* transgene of the *6b4* locus, which binds JMJ14, endogenous genes that bind JMJ14 do not appear particularly prone to producing endogenous siRNAs when RQC is compromised. Altogether with our results, these results reinforce the idea that transgene DNA hypermethylation caused by the absence of JMJ14 is specific to transgenes and not endogenous genes.

We not only found that transgenes and endogenous genes exhibit different epigenetic behavior with regard to the H3K4me3 demethylase JMJ14, but also to the histone H3K9me2 demethylase IBM1[33–35,47]. Indeed, in *jmj14* mutant plants, an increase in DNA methylation is observed in the newly inserted sequences of

the *6b4* locus but not in endogenous genes, whereas in *ibm1* mutant plants, an increase in DNA methylation is observed in endogenous genes but not in the newly inserted sequences of the *6b4* locus (Figs. 5 and 6 and Supplementary Fig. 3). Moreover, whereas *6b4* plants do not trigger PTGS spontaneously but undergo systemic PTGS when grafted on silenced rootstocks, *6b4 jmj14* plants do not trigger PTGS spontaneously and are incapable of undergoing systemic PTGS, while *6b4 ibm1* plants trigger PTGS spontaneously, similar to *6b4 drm2 cmt3* plants (Table 1 and Supplementary Fig. 5), indicating that JMJ14 and IBM1 have opposite effects on transgene PTGS. It may be assumed that actively expressed transgenes behave epigenetically the same as endogenous genes. However, our data suggest that despite being stably expressed in a WT background, transgenes exhibit epigenetic features that strongly differs from that of endogenous genes with regard to JMJ14 and IBM1 regulation.

To conclude, our research shows that JMJ14 maintains expressed transgenes in an epigenetic state that allows a residual level of aberrant RNA to be produced from the transgene body, thereby rendering the transgene particularly susceptible to PTGS. This epigenetic phenomenon could therefore represent an evolutionary probation period for the expressed transgene, until it is either completely silenced, or alternatively, epigenetically accepted as an endogenous locus in the new host genome.

## Methods

**Plant material**. The transgenic reporter lines *10027-3, 214, L1, 6b4* and *6b4-306*, and the mutants *jmj14-4, nac52^{sgs1}, nac50 nac52, rdr6^{sgs2-1}, cmt3-7, drm2-3, ibm1-1, vcs-9* all are in the *Arabidopsis* accession Columbia[13,21,26,38,40,44,47,61,62]. The transgenic lines carrying the tagged constructs *pJMJ14:JMJ14-Myc, pJMJ14:Flag-JMJ14* and *pUBQ10:NAC52-GFP* also are in the *Arabidopsis* accession Columbia[28,43,46]. Primers used for genotyping are listed in Supplementary Table 1.

**Growth conditions and grafting techniques**. *Arabidopsis* seeds were surface-sterilized, sown on a nutritive medium (1.3% S-medium Duchefa, 1% Phytoblend agar), vernalized at 4 °C for 2 days, and then placed in a culture chamber at 23 °C, 70% humidity, 120 µE m$^{-2}$ light with a 16 h light/8 h dark (long-days) or 8 h light/16 h dark (short-days) photoperiod. Seedlings grown under long-day conditions were transferred to soil after 2 weeks. Seedlings grown under short-day conditions were used for grafting experiments as described in ref. [63]. Briefly, 6 days after gemination, seedlings were cut transversely across the hypocotyl using a razor blade (90° butt graft). Then, scions and rootstocks were placed on a nitrocellulose filter (Hybond). Hypocotyls of scions and rootstock were introduced into a silicon microtube (2 mm long) to connect them to each other, and incubated under short-day conditions (8 h light, 16 h dark) for 7 to 14 days. Grafted seedlings that did not show adventitious roots were transferred to soil and grown under a short-days photoperiod.

**GUS activity and GUS RNA analysis**. GUS protein was extracted from plant leaves, and GUS activity was quantified by monitoring the quantity of 4-methylumbelliferone produced by cleavage of the substrate 4-methylumbelliferyl-b-D-glucuronide (Duchefa) on a fluorometer (Thermo Scientific fluoroskan ascent)[64].

RNA extraction and HMW or LMW RNA gel blot analyses were performed using 5–10 µg of total RNA and *GUS, U6* and *25S* probes[16]. For the reverse transcription, RNA was treated with DNaseI (Invitrogen) and 1 µg of DNA-free RNA was reverse transcribed with the primer called RT_ASGUS_Linker using the RevertAid H Minus Reverse Transcriptase (ThermoFisher, http://www.thermofisher.com/) Amplification was performed by using the LK and RbcS1Rev primers[11], and qPCR results were normalized with *EiF1a*[11].

**Whole-genome bisulfite sequencing and DNA methylation analysis**. Plants were grown under short-day conditions for five weeks (experiment 1: *6b4-1, 6b4 jmj14, 6b4 jmj14 cmt3 drm2, 6b4 cmt3 drm2*) or eight weeks (experiment 2: *6b4-2, 6b4 nac50 nac52, 6b4 ibm1*). Genomic DNA was isolated from leaf tissue from a mix of eight plants per genotype using the Nucleospin Plant II Maxi kit (Macharey Nagel), and library preparation and sequencing was performed by BGI Genomics (Hong Kong). Briefly, genomic DNA was fragmented to 100–300 bp by sonication, end-repaired, and ligated to methylated adaptors. Bisulfite treatment was then performed using the EZ DNA Methylation-Gold kit (Zymo), and the bisulfite-treated fragments were PCR amplified and sequenced as paired-end 100 bp reads (PE100) with DNBSEQ technology. Pre-processed high-quality reads were mapped to the TAIR10 genome using bismark with default settings for paired-end

libraries[65], and all downstream analysis were performed using custom R scripts. All figures presenting bisulfite data were generated using R version 4.0.2. A summary of all bisulfite sequencing data generated in this study is presented in Supplementary Table 2, and is accessible through NCBI's Gene Expression Omnibus (GSE152584).

**ChIP-qPCR analysis**. ChIP was performed on chromatin using 2 g of crosslinked in vitro shoots from 15-day-old seedlings[66]. Chromatin was sonicated (30 s ''ON''/30 s ''OFF'') with a Bioruptor UCD200 (Diagenode). The chromatin solution was diluted 10 fold with ChIP dilution buffer. Fifty microliters of Dynabeads Protein G (invitrogen) and 25 µL of GFP-Trap Dynabeads (chromotek) was washed twice with ChIP dilution buffer. Nineteen micrograms of Flag (Sigma F3165) antibodies were added to G Protein and incubated at least 2 h at 4 °C with gentle rotation. Beads were washed three times with ChIP dilution buffer. Then, 1 mL of the diluted chromatin was added to the beads/antibodies and incubated overnight at 4 °C with gentle agitation. Beads were washed as decribed[66]. After the last TE wash, reverse crosslinking (at least 4 h at 65 °C) and elution were performed using an IPure kit (Diagenode (AL-100-0100)). The final elution was performed in 60 µL and the chromatin was stored at −20 °C until analysis.

Using the Biorad-CFX-Maestro Software, the ChIP was analyzed by qPCR using 2 µL of chromatin in triplicate. Primers are listed in Supplementary Table 1. The mean of three qPCR results (with SD < 0.4 cycle threshold) was used for each point. Glyceraldehyde-3-phosphate dehydrogenase (GAPDH) was used as an internal ref. [67]. Results are represented as fold change: normalized expression (ΔΔCq) given by the ratio of Relative Quantity of the sample (2$^{(Cq\ 6b4-Cq\ sample)}$ for each identical oligo with 100% efficiency) divided by the Relative Quantity of GAPDH. At least two biological replicates were analyzed each time. Results show the mean and SD of the independent biological replicates.

**Co-immunoprecipitation experiment**. For each bulk of plants 15 days after germination, 2 g of fresh tissue was ground in CoIP Buffer (50 mM Tris pH 7.5, 100 mM EDTA, 15% Glycerol, 1% NP40, 1% Triton X100, 1x cOmplete™ Protease Inhibitor Cocktail (Merck). After 30 min on ice, samples were centrifuged at 7650 × g at 4 °C and the supernatant quantified after miracloth filtration. Five micrograms of this crude extract was submitted to preclear on 50 µL of magnetic beads coated with protein A for 2 h at 4 °C. After magnetic decantation, 50 µL of GFP-Trap_M (gtm-20, Chromotek) was rinsed (2x PBS0.1% and 1x CoIP buffer) and added to the supernatant. This was incubated overnight at 4 °C with gentle agitation. The GFP beads were then rinsed 6x in CoIP buffer, and finally adjusted to 40 µL of CoIP buffer, 5 µL of 6x Laemmli buffer and incubated at 95 °C for 5 min. After vortexing, 40 µL of this final supernatant was loaded onto a 10% polyacrylamide gel. Following sodium dodecyl sulphate–polyacrylamide gel electrophoresis, western blotting was performed using anti myc (mouse 9E10, sigma-M4439), anti GFP (Rabbit Abcam-Ab290) or anti tubulin (mouse B-5-1-2, sigma-T5168) antibodies. An ImageQuant-LAS4000 (GE Healthcare) was used for detection.

**Reporting summary**. Further information on research design is available in the Nature Research Reporting Summary linked to this article.

## Data availability
Data supporting the findings of this work are available within the paper and its Supplementary Information files. A reporting summary for this Article is available as a Supplementary Information file. The datasets and plant materials generated and analyzed during the current study are available from the corresponding author upon request. Bisulfite sequencing data are accessible through NCBI's Gene Expression Omnibus GSE152584. The source data underlying Figs. 1A–C, 2B, and 4C, as well as Supplementary Figs. 4 and 5 are provided as a Source Data file. Source data are provided with this paper.

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

## Acknowledgements

We thank Dr. Angélique Déléris for providing the *pJMJ14:JMJ14-Myc* line, Dr. Yoo-Sun Noh for providing the *pJMJ14:Flag-JMJ14* line, and Dr. Xin-Jian He for providing the *nac50 nac52* mutant.

## Author contributions

N.B., A.Y., I.L.M., C.T., N.R.G., J.C., and S.B. conducted the experiments; F.B., T.E., A.S., B.J.C., and H.V. analyzed the data; B.J.C. and H.V. designed the experiments and wrote the paper.

## Competing interests

The authors declare no competing interests.
