## [Peer Review File · Nature Communications]

REVIEWER COMMENTS

Reviewer #1 (Remarks to the Author):

In the manuscript titled "Contrasting epigenetic control of transgenes and endogenous genes promotes post-transcriptional transgene silencing in Arabidopsis", the authors demonstrate that JMJ14 participates in demethylation of strong promoters driving transgene expression in order to maintain silencing of these foreign genetic elements. This is a follow-up study from a previously published studies in 2012 and 2017 about the impact of this silencing factor and its co-factors in transgene silencing. In this study, the group adds that the demethylation and higher levels of transcription from these transgenic insertions are also needed for graft transmissible silencing, which is a nice finding but not a ground-breaking new discovery. Overall, this work is a logical, incremental advance on previous work on these factors as well as the NAC co-factors of this JMJ protein. Thus, I am not convinced it that it should be published in Nature Communications, but I leave that final determination to the Editors of the journal. I do however believe that this study also suffers from a major deficiency, which is that they researchers constantly talk about the aberrant transcripts coming from the transgenes and the effects of JMJ14 and its co-factors on the levels of these aberrant transcripts. However, there is no experimental evidence to support this assertion provided anywhere in the manuscript. The Authors need to provide experimental evidence of aberrant transcript levels for WT as compared to all silencing co-factors used throughout this work. Otherwise, once again they really can only conclude about the overall level of RNAs coming from the GUS transgene locus and not discuss aberrant transcripts without measuring levels of these RNAs. Thus, I lay out all necessary major experimental and editorial revisions needed below.

Necessary experimental revisions:

- 1) An experimental measure of aberrant transcripts from the transgenic loci (e.g. GUS) need to be added to this manuscript for WT as compared to all of the mutant variants also described. Without this experiment the Authors need to change their nomenclature to describe overall RNA levels, which have been looked at via Northern and small RNA blots. It might be higher levels of aberrant RNAs leading to the silencing, but without direct evidence of this the Authors cannot discuss this idea without direct proof.
- 2) In all quantitative RT-PCR and ChIP-PCR analyses supporting statistical analysis needs to be done and provided in the manuscript. Bars without significant testing are very uninformative to readers.

Necessary editorial revisions:

- 1) The Discussion is extremely long and fully of conjecture given the lack of experimental evidence presented in this manuscript (i.e. no direct evidence of changing aberrant RNA levels in the various mutant lines). This section needs to be significantly shortened and needs to describe only the actual findings presented in the manuscript. The description of their overall model also goes on for 2-3 pages and needs to be significantly shortened and streamlined once again focusing only on the actual findings presented in the study.

Reviewer #2 (Remarks to the Author):

In this manuscript Vaucheret group reports control of DNA methylation in transgenes and endogenous genes and its connection to post-transcriptional gene silencing (PTGS) in Arabidopsis mutants of histone demethylase genes JMJ14. A previous report by Baulcombe group showed that mutations of JMJ14 gene compromise mobile RNA silencing and induce CH methylation increase in the promoter of a transgene (Searle et al 2010). After that, similar effects were also reported by Vaucheret and collaborators (Le Masson et al 2012; Butel et al 2017). In Butel et al 2017, they showed that PTGS defect in *jmj14* is suppressed by mutation of DNA methylase gene DRM2, suggesting that the effect of

jmj14 on PTGS is mediated by increased DNA methylation. Here this group identified new alleles of jmj14 mutation from genetic screen to affect transmission of PTGS from root to shoot (Fig 1). Consistent with previous papers, the new jmj14 mutants showed increased CH methylation in the transgene (Figure 4, 5), and the PTGS defect of jmj14 was suppressed by mutations of DNA methylases (Table1). They also compared the effects of the mutants of JMJ14 to that of another histone demethylase gene IBM1/JMJ25. While the ibm1 mutation affects DNA methylation in endogenous genes, jmj14 did not affect endogenous genes but only affected the transgene.

According to the title and Summary, the main topic of the manuscript seems difference of epigenetic controls between transgenes and endogenous genes, as well as the effect of the chromatin control on PTGS. Effects of JMJ14 H3K4 demethylase on DNA methylation and chromatin are potentially interesting, but in this manuscript, I could not find much progress in regard to the pathway from JMJ14 to DNA methylation, or molecular basis for the difference in controls between transgenes and endogenous genes. Most results shown are confirmation of previous reports. Below are specific comments and suggestions, which might be useful for the authors to improve the manuscript.

1. In Summary line 28-30, "the specific attenuation of transgene DNA methylation enhances transcription of aberrant RNAs that readily induce systemic post-transcriptional transgene silencing (PTGS)." The authors showed that cmt3 drm2 mutation suppressed the effect of jmj14 on transmissible PTGS (Table 1), consistent with their previous report (Butel et 2017) that jmj14 mutation affects PTGS via DNA methylation. I could not find results showing that "transcription of aberrant RNA" is affected by DNA methylation. I wonder if they could directly see effect of cmt3 drm2 on aberrant RNAs in the background jmj14 mutant. I would suggest that at least they should examine the effect of cmt3 drm2 on the loss of siRNA in jmj14 mutant (Fig 3B).

2. The jmj14 mutation is reported to show change in DNA methylation for some of endogenous sequences, such as AtSN1 and solo LTR (Searle et al 2010). Assuming that one of the main topic of the submitted manuscript is difference between the transgene and endogenous genes, I would suggest to examine endogenous target of JMJ14 more extensively using genome-wide DNA methylation data of the mutants in the submitted manuscript.

3. As primary molecular function of JMJ14 on chromatin is likely to be on H3K4 methylation, effects of jmj14 mutation on H3K4 trimethylation could be examined genome-wide using ChIP-seq. H3K4 methylation might behave in different manners in transgenes and endogenous genes.

4. Examining localization of JMJ14 and NAC52 proteins genome-wide might be informative. As they have already made GFP-tagged lines for these proteins, ChIP-seq would be feasible. Again, genome-wide comparison might give some insights in regard to the unique properties of transgene.

5. Overall organization of the manuscript does not seem consistent. Summary is started by the sentence "Transgenes that are stably expressed in plant genomes over many generations have been assumed to behave epigenetically the same as endogenous genes", and the Summary is ended by "..., suggesting that it takes many generations for a host genome to assimilate expressed transgenes and accept them epigenetically as endogenous sequences." Thus, I expected that the results describe behavior of the transgene over generations. In the main text, however, the term "generation" and "assimilate" is used only once, and that is in the last sentence of Discussion. It is unclear what authors meant by "many generations" and "assimilate". It is not clear either what is the scientific basis for the assumption in the first sentence of Summary. If there is a literatures supporting that assumption, that should be cited.

6. Another inconsistency of the manuscript is about introduction of IBM1/JMJ25. Main topic described in Summary is comparison of transgene and endogenous genes in regard to the response to mutations in JMJ14 and IBM1/JMJ25 demethylases. Strangely, however, nothing is written about IBM1/JMJ25 in Introduction. The first time the term "IBM1" appeared in the main text is at the end of Results, where

the authors described their genome-wide DNA methylation data with citing Saze et al (2008). The cited paper is the first paper on *ibm1* and only describes effect of *ibm1* mutation on just one locus. It seems strange to me that they did not cite multiple following papers from multiple labs, that describe genome-wide analyses of DNA methylation in the *ibm1* mutants.

<https://pubmed.ncbi.nlm.nih.gov/19262562/>

<https://pubmed.ncbi.nlm.nih.gov/20834229/>

<https://pubmed.ncbi.nlm.nih.gov/24003136/>

Citing these papers appropriately would be important in the context of the submitted paper, because *ibm1* does not affect all endogenous genes; many of endogenous genes are unaffected by *ibm1*. Thus, it is not surprising that the transgene they examined was unaffected by *ibm1* mutation. If the authors of the submitted manuscript intentionally neglected these previous literatures, expecting that reviewers and readers might regard their presentation of genome-wide *ibm1* effect on endogenous gene as original of the submitted manuscript, that would be inappropriate ethically. In any case, previous works should be cited in more objective manners, to avoid to mislead the readers.

Reviewer #3 (Remarks to the Author):

In the manuscript "Contrasting epigenetic control of transgenes and endogenous genes promotes post transcriptional transgene silencing in Arabidopsis", the authors recovered three *jmj14* mutants through a forward genetics screen using the transgenic line 10027-3 (with a p35S:GFP transgene). They all showed defects in root-to-shoot PTGS transmission. To further study the molecular function of JMJ14 in transgene PTGS, a two-component GUS silencing system, 6b4-306, was used. They found that JMJ14 can interact with the p35S:GUS transgene, and *jmj14* mutation promotes CHH and CHG methylation of transgene sequences, which reduces the transcription of aberrant RNAs and thus blocking aberrant RNA-induced PTGS.

In general, the data provided by the authors are of interest and value to the community. Several points need to be clarified:

1, JMJ14 is a H3K4me3 demethylase. *jmj14* mutations are expected to cause an increase of H3K4me3. However, a previous publication by the authors' group reported a decrease of H3K4m3 in the transgene (Le Masson et al., 2012) and they stated: "Removing JMJ14 likely allows other H3K4 demethylases encoded by the Arabidopsis thaliana genome to act on transgenes and reduce transcription levels, thus preventing the triggering of S-PTGS."

The authors should use public RNA-Seq data to explore/exclude other possibilities: is it possible that H3K4me3 methyltransferase(s) acting on the transgenes may have reduced expression in *jmj14* mutants? Other factors which contribute to the H3K4m3 level on the transgene could also be differentially expressed between *jmj14* and WT [Cattaneo et al. reported that 1526 genes found to be differentially expressed between *jmj14* and Col-0 (Cattaneo et al. 2019, Development)].

2, There is a decrease of DNA methylation in the transgenes in *ibm1* mutants. How do *ibm1* mutations cause a reduction in DNA methylation at the transgenes?

3, In the manuscript, the authors stated several times that no increase in CHH and CHG methylation was observed in the endogenous genomic loci in *jmj14* and *nac50nac52* mutants:

"WGBS of 6b4 WT and 6b4 *nac50 nac52* plants revealed that no endogenous loci behave like the 6b4 locus in response to loss of NAC50 and NAC52. Indeed, no increase in CHH and CHG methylation was observed in 6b4 *nac50 nac52* plants except at the 6b4 locus (Figures 4B and 5)"

"An increase in CHG methylation was particularly evident in the promoter and gene body of the p35S:GUS transgene, and also parts of the integrated Agrobacterium vector sequences (Figure 4 and Figure S3), whereas no change in DNA methylation profiles was observed at any endogenous genomic loci in these mutants (Figure 5)."

"Remarkably, WGBS of 6b4 WT and 6b4 *jmj14-4* plants revealed that no endogenous loci behave like the 6b4 locus in response to loss of JMJ14 (Figure 5). Indeed, no increase in CHH and CHG methylation was observed in 6b4 *jmj14-4* plants except at the 6b4 locus (Figures 4 and S3A), indicating that the epigenetic behavior of transgenes strongly differs from that of endogenous genes."

However, meta plots (Fig. 5) can only show the overall pattern in genome-wide scale, but cannot display possible locus-specific differences. It is known that mutations of JMJ14 can cause a partial reduction of DRM2-dependent RdDM, and 223 DMRs were identified in *jmj14* mutants (Greenberg et al. 2013, PLOS Genetics).

4, The authors concluded that transgenes and endogenous genes exhibit different epigenetic behavior with regard to JMJ14 and IBM1. This is a very important conclusion. What are the differences in epigenetic marks (DNA methylation, histone marks or others) that may explain this differential epigenetic behavior?

5, The authors found that "Together, these results strongly suggest that JMJ14 binds to chromatin at the promoter of the p35S:GUS transgene, which promotes the binding of NAC50/NAC52 to the 35S promoter, but particularly to the downstream GUS coding sequence". JMJ14 forms a complex with NAC50 and NAC52, so why does JMJ14 binds to the chromatin at the 35S promoter but NAC50 and NAC52 preferentially bind to the downstream GUS coding sequence?

Reviewer #1 (Remarks to the Author):

Necessary experimental revisions:

1) An experimental measure of aberrant transcripts from the transgenic loci (e.g. GUS) need to be added to this manuscript for WT as compared to all of the mutant variants also described. Without this experiment the Authors need to change their nomenclature to describe overall RNA levels, which have been looked at via Northern and small RNA blots. It might be higher levels of aberrant RNAs leading to the silencing, but without direct evidence of this the Authors cannot discuss this idea without direct proof.

Answer: We agree with reviewer#1 that quantifying aberrant RNAs was absolutely necessary to support our hypothesis. We previously identified an uncapped antisense RNA (referred to as abSUG), which is not the RDR6-derived copy of a siRNA-guided GUS mRNA cleavage product because it is detected in a *rdr6* background. The amount of this uncapped abSUG RNA directly correlates with the capacity of the 35S:GUS locus to trigger RDR6-dependent PTGS. Indeed, abSUG RNA is detected in L1 *rdr6* plants (the L1 wild-type line triggers RDR6-dependent PTGS with 100% efficiency) and is only barely detected in 6b4 *rdr6* plants (the 6b4 wild-type line never triggers RDR6-dependent PTGS spontaneously). However, it is detected in 6b4 *xrn3 xrn4 rdr6* plants (the 6b4 *xrn3 xrn4* line triggers RDR6-dependent PTGS as efficiently as the L1 wild-type line). Therefore, we hypothesized that, when it is not degraded by XRN enzymes, this uncapped abRNA is transformed into dsRNA by RDR6 to trigger RDR6-dependent PTGS (Parent et al, 2015, Nuc. Acids Res.). In the revised manuscript, we show that the amount of this uncapped abSUG is lower in L1 *jmj14 rdr6* plants compared to L1 *rdr6* (Figure 3), supporting our hypothesis that JMJ14 promotes the production of abRNAs that are required for the triggering of RDR6-dependent PTGS.

2) In all quantitative RT-PCR and ChIP-PCR analyses supporting statistical analysis needs to be done and provided in the manuscript. Bars without significant testing are very uninformative to readers.

Answer: Statistical analyses are now provided in the revised manuscript.

Necessary editorial revisions:

1) The Discussion is extremely long and fully of conjecture given the lack of experimental evidence presented in this manuscript (i.e. no direct evidence of changing aberrant RNA levels in the various mutant lines). This section needs to be significantly shortened and needs to describe only the actual findings presented in the manuscript. The description of their overall model also goes on for 2-3 pages and needs to be significantly shortened and streamlined once again focusing only on the actual findings presented in the study.

Answer: We have shortened the discussion and removed the model and its long description.

Reviewer #2 (Remarks to the Author):

*1. In Summary line 28-30, “the specific attenuation of transgene DNA methylation enhances transcription of aberrant RNAs that readily induce systemic post-transcriptional transgene silencing (PTGS).” The authors showed that *cmt3 drm2* mutation suppressed the effect of *jmj14* on transmissible PTGS (Table 1), consistent with their previous report (Butel et 2017)*

that jmj14 mutation affects PTGS via DNA methylation. I could not find results showing that “transcription of aberrant RNA” is affected by DNA methylation. I wonder if they could directly see effect of cmt3 drm2 on aberrant RNAs in the background jmj14 mutant. I would suggest that at least they should examine the effect of cmt3 drm2 on the loss of siRNA in jmj14 mutant (Fig 3B).

Answer: We agree with reviewer#2 that quantifying aberrant RNAs was absolutely necessary to support our hypothesis. We previously identified an uncapped antisense RNA (referred to as abSUG), which is not the RDR6-derived copy of a siRNA-guided GUS mRNA cleavage product because it is detected in a rdr6 background. The amount of this uncapped abSUG RNA directly correlates with the capacity of the 35S:GUS locus to trigger RDR6-dependent PTGS. Indeed, abSUG RNA is detected in L1 rdr6 plants (the L1 wild-type line triggers RDR6-dependent PTGS with 100% efficiency) and is only barely detected in 6b4 rdr6 plants (the 6b4 wild-type line never triggers RDR6-dependent PTGS spontaneously). However, it is detected in 6b4 xrn3 xrn4 rdr6 plants (the 6b4 xrn3 xrn4 line triggers RDR6-dependent PTGS as efficiently as the L1 wild-type line). Therefore, we hypothesized that, when it is not degraded by XRN enzymes, this uncapped abRNA is transformed into dsRNA by RDR6 to trigger RDR6-dependent PTGS (Parent et al, 2015, Nuc. Acids Res.). In the revised manuscript, we show that the amount of this uncapped abSUG is lower in L1 jmj14 rdr6 plants compared to L1 rdr6 (Figure 3), supporting our hypothesis that JMJ14 promotes the production of abRNAs that are required for the triggering of RDR6-dependent PTGS.

2. The jmj14 mutation is reported to show change in DNA methylation for some of endogenous sequences, such as AtSN1 and solo LTR (Searle et al 2010). Assuming that one of the main topic of the submitted manuscript is difference between the transgene and endogenous genes, I would suggest to examine endogenous target of JMJ14 more extensively using genome-wide DNA methylation data of the mutants in the submitted manuscript.

Answer: We acknowledge that Searle et al (2010) used methylation-sensitive PCR to report evidence of a decrease in DNA methylation at an AluI site(s) of a AtSN1 retrotransposon(s) and a MspI site of a soloLTR retrotransposon(s) of jmj14 mutants. However, this decreased methylation was not quantified and the authors did not examine the extent of methylation at all CG, CHG and CHH sites within these retrotransposon loci. In contrast, we have performed DNA methylation analysis genome-wide. A summary of all DMRs identified is now presented in Table S3. Only one endogenous hyper CHG DMR was identified in jmj14 and only one endogenous hyper CHG DMR was identified in nac50nac52; however, these two DMRs were on chromosome 5 and chromosome 3, respectively, indicating that jmj14- and nac50nac52-dependent hyper CHG methylation is only found in the transgenic sequences (both the T-DNA and the co-integrated backbone sequences of the binary vector). Moreover, transgenic sequences not only exhibit hyper CHG methylation but also hyper CG and hyper CHH methylation (Fig 6 and Fig S3). The only endogenous hyper CHG methylation found in jmj14 did not show hyper CG and hyper CHH methylation, while the only endogenous hyper CHG methylation found in nac50nac52 showed hyper CHH methylation but not hyper CG methylation. Therefore, it is most likely that the endogenous DMRs observed between wild type and either jmj14 or nac50nac52 correspond to endogenous genomic sequences that exhibit highly variable methylation level (Becker et al, 2011, Schmitz et al, 2011).

3. As primary molecular function of JMJ14 on chromatin is likely to be on H3K4 methylation, effects of jmj14 mutation on H3K4 trimethylation could be examined genome-wide using

ChIP-seq. H3K4 methylation might behave in different manners in transgenes and endogenous genes.

Answer: The analysis requested by Reviewer 2 was reported in our previous paper (Le Masson et al, 2012). In this publication, we confirmed a genomewide increase in H3K4me3 level in endogenous sequences of the *jmj14* mutant (as expected when mutating a H3K4me3 demethylase) but also reported a decrease of H3K4me3 level in the transgenic sequences of the *jmj14* mutant (as determined by ChIP).

4. Examining localization of JMJ14 and NAC52 proteins genome-wide might be informative. As they have already made GFP-tagged lines for these proteins, ChIP-seq would be feasible. Again, genome-wide comparison might give some insights in regard to the unique properties of transgene.

Answer: The endogenous genome binding sites of JMJ14 and NAC52 have been determined by others (Ning et al, 2015, Zhang et al, 2015), and we show in this paper that the 35S:GUS transgene also binds JMJ14 and NAC52. However, the simultaneous increase of DNA methylation at CG, CHG and CHH sites in *jmj14* and *nac50nac52* background is only observed at transgenic sequences, not at endogenous genomic sequences.

5. Overall organization of the manuscript does not seem consistent. Summary is started by the sentence “Transgenes that are stably expressed in plant genomes over many generations have been assumed to behave epigenetically the same as endogenous genes”, and the Summary is ended by “..., suggesting that it takes many generations for a host genome to assimilate expressed transgenes and accept them epigenetically as endogenous sequences.” Thus, I expected that the results describe behavior of the transgene over generations. In the main text, however, the term “generation” and “assimilate” is used only once, and that is in the last sentence of Discussion. It is unclear what authors meant by “many generations” and “assimilate”. It is not clear either what is the scientific basis for the assumption in the first sentence of Summary. If there is a literature supporting that assumption, that should be cited.

Answer: The last sentence of our abstract is a hypothesis. It is well beyond the scope of this manuscript to determine how many generations are required to assimilate a foreign exogenous DNA sequence such that it is epigenetically indistinguishable from endogenous sequences. It could take more than a human life time and may require epigenetic reprogramming of the genome following a severe environmental or genetic stress event, for example. The 6b4 and L1 transgenic lines were produced in 1994 and propagated since then, and the transgenes still respond differently to endogenous sequences when in *jmj14* and *nac52* mutant genetic backgrounds. We have added this information to the discussion. We have changed the first sentence of the abstract to: “Transgenes that are stably expressed in plant genomes over many generations could be assumed to behave epigenetically the same as endogenous genes.”

*6. Another inconsistency of the manuscript is about introduction of IBM1/JMJ25. Main topic described in Summary is comparison of transgene and endogenous genes in regard to the response to mutations in JMJ14 and IBM1/JMJ25 demethylases. Strangely, however, nothing is written about IBM1/JMJ25 in Introduction. The first time the term “IBM1” appeared in the main text is at the end of Results, where the authors described their genome-wide DNA methylation data with citing Saze et al (2008). The cited paper is the first paper on *ibm1* and only describes effect of *ibm1* mutation on just one locus. It seems strange to me that they did not cite multiple following papers from multiple labs, that describe genome-wide analyses of*

DNA methylation in the ibm1 mutants.
<https://pubmed.ncbi.nlm.nih.gov/19262562/>
<https://pubmed.ncbi.nlm.nih.gov/20834229/>
<https://pubmed.ncbi.nlm.nih.gov/24003136/>

Citing these papers appropriately would be important in the context of the submitted paper, because ibm1 does not affect all endogenous genes; many of endogenous genes are unaffected by ibm1. Thus, it is not surprising that the transgene they examined was unaffected by ibm1 mutation. If the authors of the submitted manuscript intentionally neglected these previous literatures, expecting that reviewers and readers might regard their presentation of genome-wide ibm1 effect on endogenous gene as original of the submitted manuscript, that would be inappropriate ethically. In any case, previous works should be cited in more objective manners, to avoid to mislead the readers.

Answer: We apologize for not citing all the relevant ibm1 literature. It was not our intention to neglect this literature and imply that our report of genome-wide hypermethylation in ibm1 was novel. Indeed, in the original submission, we wrote that “WGBS analysis confirmed increased CHH and CHG methylation in the body of endogenous genes”, but forgot to include the corresponding citations. The ibm1 work is now properly cited.

Reviewer #3 (Remarks to the Author):

In general, the data provided by the authors are of interest and value to the community. Several points need to be clarified:

1, JMJI4 is a H3K4me3 demethylase. jmj14 mutations are expected to cause an increase of H3K4me3. However, a previous publication by the authors' group reported a decrease of H3K4m3 in the transgene (Le Masson et al., 2012) and they stated: "Removing JMJI4 likely allows other H3K4 demethylases encoded by the Arabidopsis thaliana genome to act on transgenes and reduce transcription levels, thus preventing the triggering of S-PTGS."

The authors should use public RNA-Seq data to explore/exclude other possibilities: is it possible that H3K4me3 methyltransferase(s) acting on the transgenes may have reduced expression in jmj14 mutants? Other factors which contribute to the H3K4m3 level on the transgene could also be differentially expressed between jmj14 and WT [Cattaneo et al. reported that 1526 genes found to be differentially expressed between jmj14 and Col-0 (Cattaneo et al. 2019, Development)].

Answer: We have mined mRNA-seq data of jmj14 and nac52 mutants. Around 400 genes are upregulated, and only five genes are downregulated, but they do not include any H3K4 methyltransferases.

2, There is a decrease of DNA methylation in the transgenes in ibm1 mutants. How do ibm1 mutations cause a reduction in DNA methylation at the transgenes?

Answer: The reduction of DNA methylation at transgenic sequences is very slight in the ibm1 mutant, and we have no explanation for this. The important ibm1 result in our manuscript is that while we confirmed an increased CHG and CHH methylation of endogenous genes in the

ibm1 mutant, the same increase in CHG and CHH methylation was not observed at the 35S:GUS transgene in the ibm1 mutant genetic background.

3, In the manuscript, the authors stated several times that no increase in CHH and CHG methylation was observed in the endogenous genomic loci in jmj14 and nac50nac52 mutants:

"WGBS of 6b4 WT and 6b4 nac50 nac52 plants revealed that no endogenous loci behave like the 6b4 locus in response to loss of NAC50 and NAC52. Indeed, no increase in CHH and CHG methylation was observed in 6b4 nac50 nac52 plants except at the 6b4 locus (Figures 4B and 5)"

"An increase in CHG methylation was particularly evident in the promoter and gene body of the p35S:GUS transgene, and also parts of the integrated Agrobacterium vector sequences (Figure 4 and Figure S3), whereas no change in DNA methylation profiles was observed at any endogenous genomic loci in these mutants (Figure 5)."

"Remarkably, WGBS of 6b4 WT and 6b4 jmj14-4 plants revealed that no endogenous loci behave like the 6b4 locus in response to loss of JMJ14 (Figure 5). Indeed, no increase in CHH and CHG methylation was observed in 6b4 jmj14-4 plants except at the 6b4 locus (Figures 4 and S3A), indicating that the epigenetic behavior of transgenes strongly differs from that of endogenous genes."

However, meta plots (Fig. 5) can only show the overall pattern in genome-wide scale, but cannot display possible locus-specific differences. It is known that mutations of JMJ14 can cause a partial reduction of DRM2-dependent RdDM, and 223 DMRs were identified in jmj14 mutants (Greenberg et al. 2013, PLOS Genetics).

Answer: We have examined DNA methylation genome-wide but acknowledge that the results were not sufficiently detailed in the original version of the manuscript. A summary of all DMRs identified is presented in Table S3. Only one endogenous hyper CHG DMR was identified in jmj14 and only one endogenous hyper CHG DMR was identified in nac50nac52; however, these two DMRs were on chromosome 5 and chromosome 3, respectively, indicating that jmj14- and nac50nac52-dependent hyper CHG methylation is only found in the transgenic sequences (both the T-DNA and the co-integrated backbone sequences of the binary vector). Moreover, transgenic sequences not only exhibit hyper CHG methylation but also hyper CG and hyper CHH methylation (Fig 4 and Fig S3). The only endogenous hyper CHG methylation found in jmj14 did not show hyper CG and hyper CHH methylation, while the only endogenous hyper CHG methylation found in nac50nac52 showed hyper CHH methylation but not hyper CG methylation. Therefore, it is most likely that the endogenous DMRs observed between wild type and either jmj14 or nac50nac52 correspond to endogenous genomic sequences that exhibit highly variable methylation level (Becker et al, 2011, Schmitz et al, 2011).

4, The authors concluded that transgenes and endogenous genes exhibit different epigenetic behavior with regard to JMJ14 and IBM1. This is a very important conclusion. What are the differences in epigenetic marks (DNA methylation, histone marks or others) that may explain this differential epigenetic behavior?

Answer: This is a very interesting question and all we can do is speculate that an epigenetic mark is laid down on naked DNA before or when it integrates into the genome. Hundreds of

chromatin modification are known and maybe more are still unknown.

5, The authors found that “Together, these results strongly suggest that JMJ14 binds to chromatin at the promoter of the p35S:GUS transgene, which promotes the binding of NAC50/NAC52 to the 35S promoter, but particularly to the downstream GUS coding sequence“. JMJ14 forms a complex with NAC50 and NAC52, so why does JMJ14 binds to the chromatin at the 35S promoter but NAC50 and NAC52 preferentially bind to the downstream GUS coding sequence?

Answer: Statistical analyses now provided in the revised manuscript show that the binding of NAC52 to the GUS coding sequence is not significant. Therefore, we have amended our conclusions, saying that both JMJ14 and NAC52 mostly bind to the 35S promoter.

REVIEWERS' COMMENTS

Reviewer #2 (Remarks to the Author):

The authors have responded to my suggestions appropriately. I do not have any further suggestions to improve the manuscript.

[Editor: Reviewer #1 and #3 privately state that they are satisfied with the revision.]